# Integration of population-based surveys for neglected tropical diseases: A scoping review

**Amanuel Belay** [1,2*], **Agumasie Semahegn**[1,2,3], **Binyam Tesfaw Hailu**[4], **Abebaw Fekadu**[1,2,5], **Gail Davey**[2,6], **Rana Ahmed**[2,7], **Hope Simpson**[2,8]

**1** Centre for Innovative Drug Development and Therapeutic Trials for Africa (CDT-Africa), College of Health Sciences, Addis Ababa University, Addis Ababa, Ethiopia, **2** Global Health and Infection Department, Brighton and Sussex Medical School, Brighton, United Kingdom, **3** College of Health and Medical Sciences, Haramaya University, Harar, Ethiopia, **4** School of Earth Sciences, Addis Ababa University, Addis Ababa, Ethiopia, **5** Department of Psychiatry and WHO Collaborating Centre in Mental Health Research and Capacity Building, School of Medicine, College of Health Sciences, Addis Ababa University, Addis Ababa, Ethiopia, **6** School of Public Health, College of Health Science, Addis Ababa University, Addis Ababa, Ethiopia, **7** Converge: Centre for Chronic Disease and Population Health Research, School of Population Health, RCSI University of Medicine and Health Sciences, Dublin, Ireland, **8** Department of Disease Control, London School of Hygiene and Tropical Medicine, London, United Kingdom

* amanbelays@gmail.com

## Abstract

### Background

Neglected Tropical Diseases (NTDs) can be controlled by interventions in populations with disease or biomarker prevalence above certain thresholds, but surveys to identify these target populations drive up the resources required. Existing survey guidelines focus on single diseases, missing opportunities to collect and analyse data on multiple co-occurring diseases more efficiently. Therefore, this scoping review aims to identify and synthesise strategies for integrating NTD surveys, evaluate costs and cost savings, and summarise key challenges and recommendations.

### Methods

We conducted a comprehensive scoping review, guided by the Joanna Briggs Institute methodology for scoping reviews and following the PRISMA-ScR guidelines. PubMed, Web of Science, and Scopus databases were systematically searched for relevant studies until September 2024. Study characteristics such as survey design, sampling methods, and survey integration approaches, challenges and recommendations were extracted and organised thematically.

### Results

A total of 2,829 relevant documents were retrieved, and 81 articles met the inclusion criteria. The use of multiplex assays, allowing simultaneous testing for multiple antibodies from a single sample, and collecting multiple sample types (blood, urine, and

**Data availability statement:** All relevant data are within the manuscript and its Supporting information files.

**Funding:** This research is funded by the National Institute for Health and Care Research (NIHR) (16/13629 to GD and AF) Global Health Research Unit Programme. The views expressed are those of the author(s) and not necessarily those of the NIHR or the Department of Health and Social Care. The funders had no role in study design, data collection and analysis, decision to publish, or preparation of the manuscript.

**Competing interests:** The authors have declared that no competing interests exist.

stool) in one visit, can save time and reduce costs. Leveraging existing platforms, such as lymphatic filariasis Transmission Assessment Surveys, malaria surveys, and the standardised trachoma prevalence surveys, also enables substantial cost savings by optimising shared resources. Integrated surveys demonstrated notable cost-efficiency and operational feasibility. However, integration can be challenging due to methodological differences, lack of coordination, community resistance, and funding challenges. Key recommendations include strong stakeholder engagement, robust planning, leveraging of existing infrastructure, community involvement, methodological flexibility, and adoption of technologies such as rapid diagnostic tests and mobile tools to optimise data collection.

## Conclusion

Integrated NTD surveys, when strategically designed and contextually adapted, enhance efficiency and reduce costs. Future initiatives should focus on optimising survey integration, leveraging existing health infrastructure, and fostering cross-program collaboration.

### Author summary

Neglected tropical diseases impact millions of people living in poverty in tropical regions, yet many of these illnesses receive limited attention and funding. Surveys play a vital role in identifying populations in need of treatment, but they are costly to carry out. In our study, we examined how health programmes can collect information for multiple diseases simultaneously instead of conducting separate surveys for each one. By reviewing existing research, we discovered that combining surveys can save time, cut costs, and make better use of limited resources. Collecting multiple samples, such as blood, urine, and stool, during a single visit or testing one sample for multiple diseases can make data collection more efficient. We also found that building on existing health programmes, such as those for malaria, trachoma, or lymphatic filariasis, helps prevent duplication of effort. However, differences in required methodologies and poor planning can complicate integration. Our findings show that successful integration demands careful planning, collaboration among partners, and strong community involvement.

## Introduction

Neglected tropical diseases (NTDs) are a group of diseases that mainly occur in tropical and sub-tropical climates and where there is inadequate access to sanitation, clean water and healthcare. As such, their highest burdens fall in remote and rural areas, informal settlements and conflict zones. NTDs affect around 2 billion

people globally and cause approximately 200,000 deaths and 25.1 million disability-adjusted life-years (DALYs) per year [1,2].

NTDs are a significant barrier to development, significantly impacting populations living in extreme poverty. Most of the global population living in extreme poverty is affected by one or more NTDs. Infection with multiple NTDs leads to irreversible disabilities and loss of productivity, resulting in socioeconomic decline [3]. Mass drug administration (MDA) can be delivered on a population level to effectively treat and prevent infection, reduce prevalence and mortality and slow transmission. Recognition of MDA as a highly cost-effective intervention, coupled with commitments from pharmaceutical companies to provide medicines freely as long as needed, led to the establishment of major control and elimination programs for lymphatic filariasis (LF), onchocerciasis, schistosomiasis, soil-transmitted helminthiases (STH), and trachoma [3]. The primary role of population-based prevalence surveys for PC-NTDs is to identify populations in need of MDA and to guide decisions about the required frequency of PC. They are also used to generate insights on disease epidemiology and burden and the impact of control programs [4–6]. For diseases targeted for elimination, prevalence surveys are an essential tool for programs to determine if the prevalence of the disease has fallen below the target threshold [7].

While being essential to ensure that resources for control are well-targeted, surveys constitute a large proportion of the costs of control and elimination programmes. Substantial resources are required for mobilising survey teams, deploying them to survey communities, and collecting, processing and analysing samples. A costing study on onchocerciasis elimination showed that the cost of treating each person for onchocerciasis in Africa could double in the later stages of elimination, mainly due to the costs of surveillance and monitoring [8]. An optimal survey design should balance accuracy with cost, providing programs with good decision-making value for money.

WHO recommends integrating control and intervention strategies, including coordinating mapping activities across NTDs and with other health programs like WASH and vaccination [5,9]. However, traditional monitoring and evaluation (M&E) survey methods focus on one disease at a time, missing the opportunity to efficiently use resources by collecting and analysing data on multiple co-occurring diseases [10]. Furthermore, in today's resource-constrained environment for global health, NTD programs are finding it increasingly difficult to identify funds for standalone M&E surveys. Integrated surveys are no longer a nice-to-have, they are now essential.

Despite the growing interest in integrating NTD surveys, a comprehensive synthesis of existing methods and strategies is lacking. Therefore, this scoping review aims to systematically identify and synthesise the existing literature on methods, strategies, challenges and opportunities for integrating NTD surveys.

## Review questions

This scoping review aims to answer the following review questions:

- Which NTDs have been co-surveyed through integrated approaches and in which countries?

- What strategies have been employed to integrate multiple NTD surveys into a single, comprehensive survey framework that is practical and cost-effective?

- What are the challenges associated with these approaches?

- What are the enablers of and barriers to successful integration of NTD surveys?

## Methods

The scoping review was conducted following the Joanna Briggs Institute (JBI) methodology [11]. The findings of the review were reported following the Preferred Reporting Items for Systematic Reviews and Meta-Analyses extension for Scoping Reviews (PRISMA-ScR) [12] (S1 PRISMA Checklist). The protocol was registered on Open Science Framework (osf.io/x4dta).

### Inclusion criteria

**Concept.** The main concept examined in this review was the integration of surveys that included at least one NTD. We included podoconiosis as an NTD, although it is not formally recognised as an NTD by WHO because it shares many characteristics of NTDs and is included within several national strategic plans on NTDs.

**Context.** The context included all settings where population-based surveys and integrated NTD surveys were conducted. This encompasses community-based house-to-house surveys and school-based surveys done in different WHO regions. The review included surveys that were done by integrating within the national health systems as well as those conducted independently by research institutions, NGOs, or international partners. There were no restrictions based on specific settings, allowing for a comprehensive understanding of the methods and strategies used globally.

### Information sources

This review considered all types of evidence sources, including primary research studies, systematic reviews, meta-analyses, guidelines, technical reports, and grey literature such as conference papers and government reports. Both quantitative and qualitative studies were included. The aim was to capture a wide range of evidence to ensure a thorough exploration of the field.

### Search strategy

We used a three-step search strategy. An initial search on PubMed was conducted and titles, abstracts and index terms of relevant studies were analysed to identify additional search terms. A second search was carried out using the additional identified search terms on MEDLINE (PubMed), Web of Science, and Scopus databases on 20th of August 2024 (S1 Appendix). Finally, a backwards search was employed to search the reference lists of all included articles for additional articles and literature that met the inclusion criteria and were included in the final review. Due to time and resource limitations, the search was limited to studies that were published in English.

### Source of evidence selection

The results of the searches from all the databases were exported to the reference manager EndNote (version 21), and duplicates were removed. Two reviewers (AB and HS) conducted pilot testing on a random sample of 100 titles/abstracts to ensure clarity and consistency in the application of the inclusion and exclusion criteria before starting the full screening process. The full screening process was done once the two reviewers achieved at least 90% agreement.

A two-stage standardised screening process using Rayyan online software was used to identify relevant articles to include in the review. Initially, an independent reviewer screened the titles and abstracts of all the identified articles based on the inclusion criteria, and a senior reviewer (HS) cross-checked. The full text of potentially relevant articles was then retrieved. An independent reviewer assessed the full-text articles, and a senior reviewer (HS) cross-checked to ensure accuracy. The screening and selection process was guided by the PRISMA-ScR flow chart [13].

### Data extraction and summary of results

The data extraction tool was adapted from the JBI methodology guidance for scoping reviews [14] (S2 Appendix). The data extraction form was pilot-tested on 20 evidence sources to ensure that appropriate and sufficient data were extracted and to assess the reliability, consistency and appropriateness of the extraction tool. Following the pilot test, one reviewer conducted the data extraction independently, and a senior reviewer (GD) cross-checked. For companion studies, where multiple publications originated from the same single survey, only the publication most relevant to the review questions was selected for data extraction to avoid duplication of evidence. Companion papers reporting secondary analyses or findings not aligned with the review objectives were identified but not extracted to avoid duplication of evidence. The extracted

data were categorised into three thematic areas: diagnosis, survey design, and costs and resource savings. We discuss how these areas translate to enabling conditions and challenges for integrated surveys.

## Results

### Study characteristics

A total of 2,829 articles were identified from the initial search, with 1473 remaining after removing duplicates (Fig 1). Of these, 129 articles were included for full text review after the initial title and abstract review and 81 articles were included in the final analysis.

The articles included were published from 2006 – 2024, with between 1 and 11 studies published each year (Fig 2).

Of the 81 studies, 67 were integrated cross-sectional surveys [15–81], five were review articles [82–86], two were meeting or consultation summary articles [87,88], two were perspective articles [89,90], one a qualitative study after an

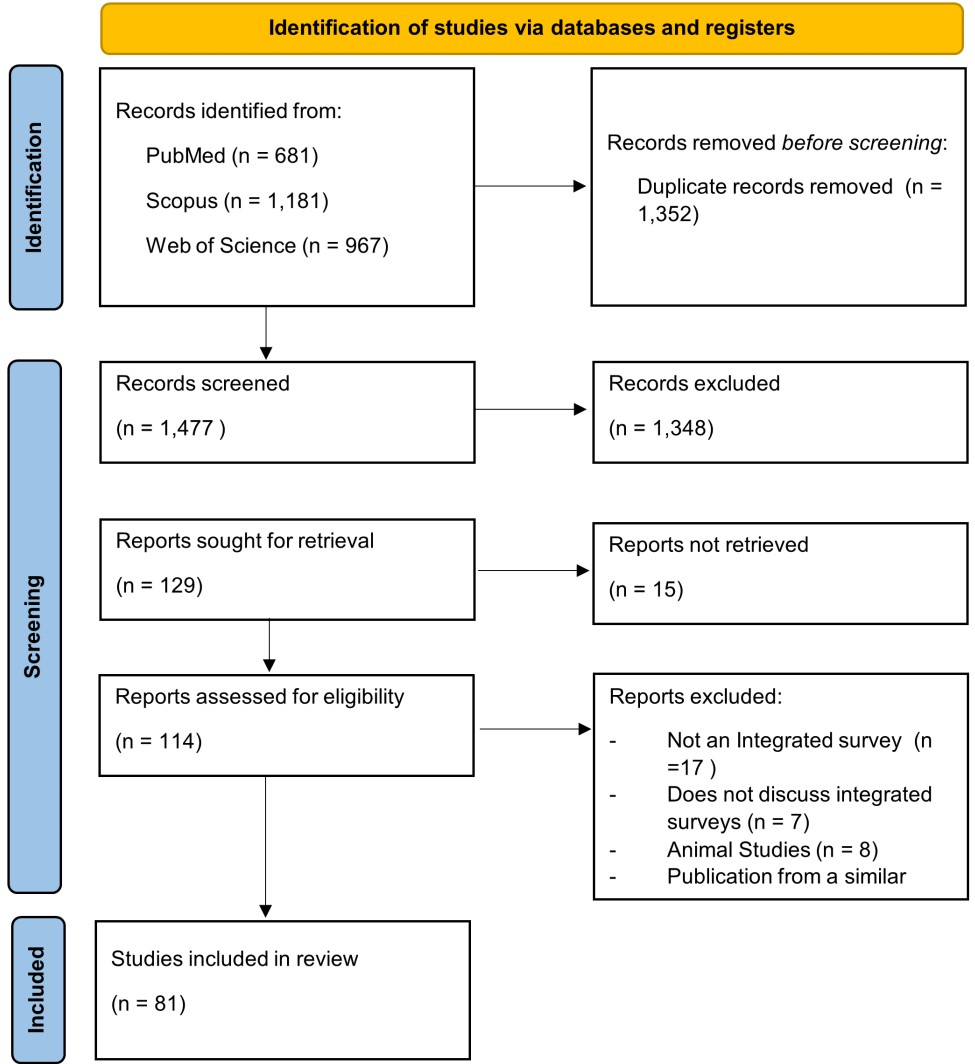

**Fig 1. PRISMA-ScR flowchart.**

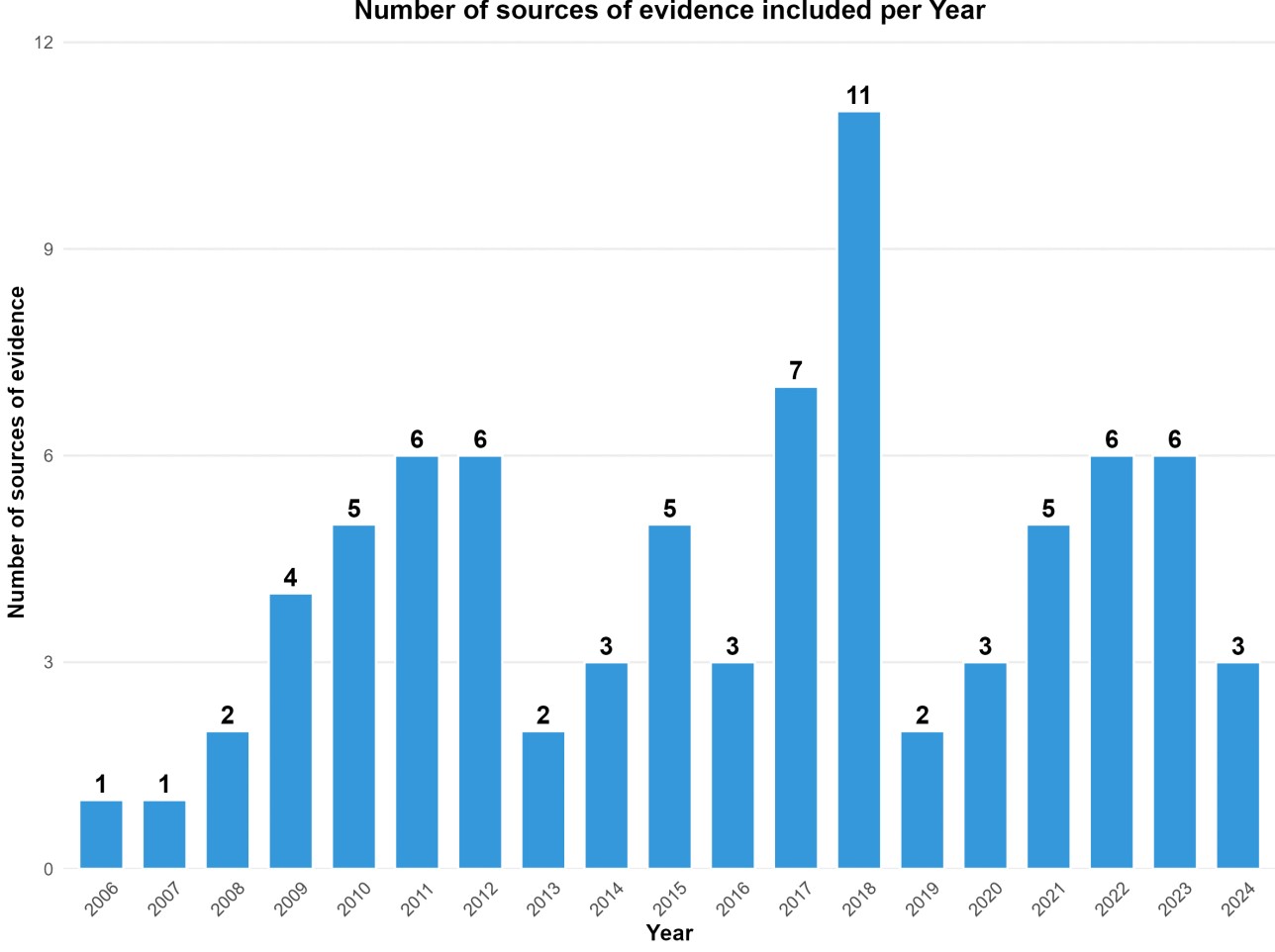

**Fig 2. Number of sources of evidence publications on integrated NTD surveys by year.**

integrated survey [91], one a qualitative document analysis of integrated surveys [92], one a "lesson from the field" perspective article [93], one an expert opinion [94], and one a guideline/roadmap document [95].

Schistosomiasis, STH, and LF were the most commonly investigated NTDs, appearing in 41, 40, and 27 studies, respectively. Malaria was identified as the most frequently integrated non-NTD, with 19 studies.

From the integrated surveys conducted, schistosomiasis and STH were the most frequently integrated diseases, with 34 surveys targeting both diseases. The next commonly integrated diseases were STH and LF, STH and malaria, and schistosomiasis and malaria, each reported in 13 studies (Fig 3).

Integrated NTD surveys were conducted in numerous countries where NTDs are endemic (Fig 4). The majority of integrated surveys (n = 57) were conducted in Africa, with the highest number in Ethiopia (n = 7), followed by Côte d'Ivoire, Kenya, Nigeria, and Tanzania (n = 5). Eight integrated surveys were conducted in Latin America (four in South America, two in Central America, and two in the Caribbean), five in Oceania and four in Asia.

Of the five identified reviews, three focused on the opportunities for the diagnosis, mapping, control and management of skin-NTDs [82,83,86]. One discussed the opportunities and challenges of integrated mapping, monitoring, and surveillance across all NTDs [85], and one review discussed challenges for an integrated rapid mapping and presented future directions for an integrated rapid mapping approach across all NTDs [84]. None of the identified reviews were systematic reviews.

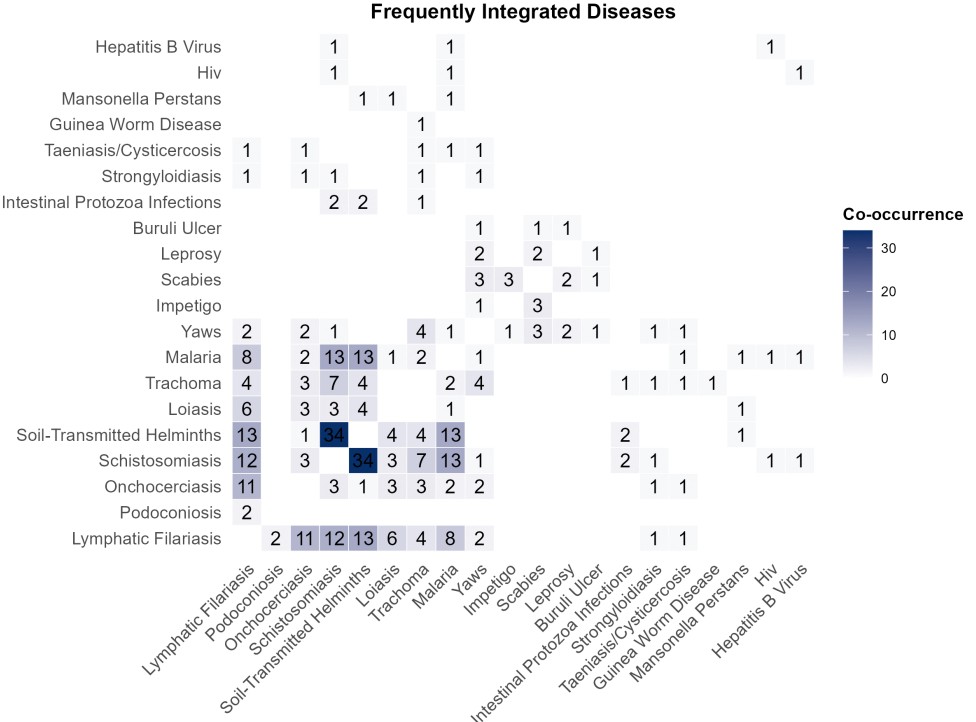

**Fig 3. Frequently integrated diseases.**

## Global Distribution of Included Integrated NTD Surveys

Count of surveys conducted per country

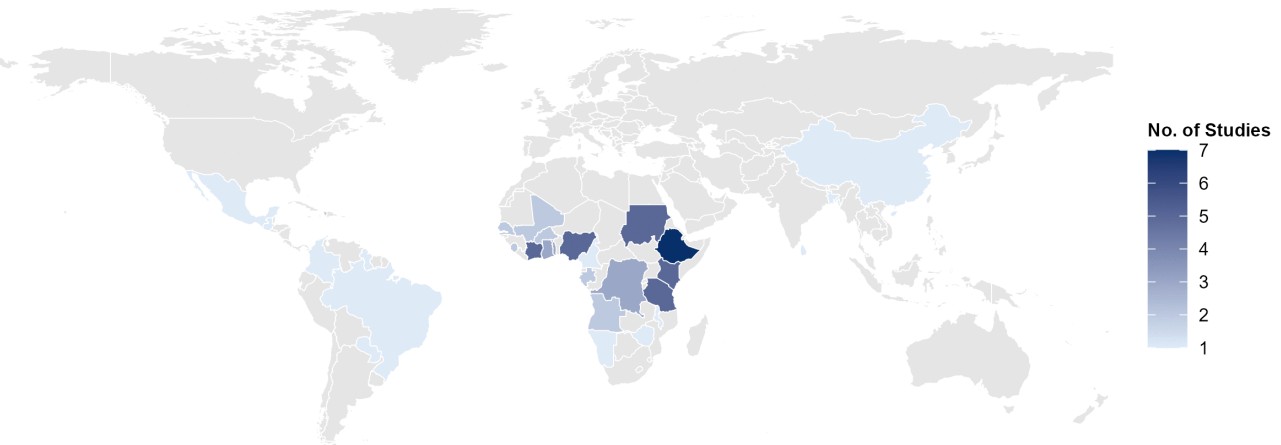

**Fig 4. Global distribution of integrated NTD surveys conducted.** Base-layer map from Natural Earth: https://www.naturalearthdata.com, licence: https://www.naturalearthdata.com/about/terms-of-use/.

## Diagnosis within integrated surveys

The simultaneous identification of multiple diseases within a single survey adds complexity beyond single-disease testing. Various methods have been applied to optimise field operations and laboratory processing, enhancing practicality and cost-effectiveness.

**Collecting multiple sample types.** The collection of different specimen types from the same individual during a single visit was used as a strategy for integrating the diagnosis of multiple diseases. This method was used in integrated surveys for diseases that require different specimens [30,68]. For example, blood, urine and stool samples were collected from each participant in a survey in Cameroon to simultaneously detect *Schistosoma (S. mansoni, S. guineensis)*, STH, and *Plasmodium (P. falciparum, P. malariae, P. ovale, and P. vivax)* [30]. Co-collecting multiple specimens from the same individual optimises survey efficiency by minimising repeated visits for sample collection, thereby reducing travel, personnel time and the overall logistics cost of the survey.

However, the use of tests and diagnostic approaches that require different specimen types adds complexity and expenses to a survey [40,45,66,84,85]. The coordination of different sample types for the different diagnostic methods is a challenge for effective integration.

**Multiplex bead assays.** The use of multiplex bead assays (MBA) was another strategy for integrating the diagnosis of multiple diseases. These assays simultaneously test antibody responses to multiple antigens from a single, small blood sample. This approach eliminates the need for a separate assay for each disease, making surveillance cost-effective. An MBA is recommended for integrated serosurveillance of diseases endemic in the same geographical area or targeting the same population [89,95]. Serological surveys (serosurveillance) measure the prevalence of antibodies against specific pathogen antigens (Ag). MBAs have been used to monitor a wide variety of NTDs, including LF, onchocerciasis, strongyloidiasis, schistosomiasis, STH, malaria, trachoma, yaws, and taeniasis/cysticercosis antibodies simultaneously from a single blood sample [24,39,92].

The development of a self-contained portable multiplex diagnostic platform capable of simultaneously testing several infections using a small number of tests is recommended as a future research direction [87]. The use of a single sample reduces the burden of sample collection, minimises transport costs, and drastically cuts down laboratory processing time per disease, making it potentially cost-effective. Furthermore, several NTDs, including onchocerciasis and trachoma, are increasingly collecting dried blood spots (DBS) as part of their routine program M&E surveys, creating opportunities to leverage MBAs to detect additional pathogens. An advantage of using DBS is that they are easy to collect, transport and store [89].

**Clinical assessment.** Many NTDs, including Buruli ulcer, cutaneous leishmaniasis, leprosy, LF, mycetoma, onchocerciasis, yaws, chromoblastomycosis and scabies, have a skin manifestation. Some NTDs, including podoconiosis and scabies, cannot be reliably identified from patient samples but present with skin manifestations. Using the skin as an entry point was suggested for integrating the assessment of multiple NTDs [83,86,88]. Diagnosis can effectively be done based on clinical signs and skin assessment without requiring laboratory tests. Using a syndrome-based assessment tool and training guides designed to enable front-line health workers to clinically diagnose NTDs based on skin changes was recommended [83]. To support this, WHO has developed "Recognising neglected tropical diseases through changes on the skin', a simple tool focusing on major skin changes such as ulcers, lumps, swollen limbs, and patches. The tool provides a diagnostic flowchart and algorithm for detecting and managing NTDs and can be used for mapping activities [83,86]. This tool is intended to be used by frontline non-specialist health workers with minimal dermatological training. The availability of simplified consensus diagnostic criteria for skin NTDs, such as scabies, onchocerciasis, and podoconiosis, facilitates and opens the opportunity for integrated mapping of skin NTDs [86]. Integrating NTDs that can be clinically diagnosed can be less time-consuming compared to those that require laboratory sample collection and processing for diagnosis, impacting the feasibility and efficiency of integration [93].

## Integrated survey methods and protocols

**Leveraging and adapting existing platforms.** Leveraging and adapting established survey platforms has been used to improve integration. The most commonly used platforms were LF Transmission Assessment Surveys (TAS) [15,35,39,44,48,50,51,53,55,72] and trachoma mapping surveys [24,41,52,57,96]. Currently, trachoma mapping

is supported by Tropical Data, a consortium of partners that supports countries' Ministries of Health to conduct epidemiologically robust, high-quality, WHO standardised prevalence surveys. In two studies, malaria surveys served as platforms to integrate surveys for trachoma [56] and schistosomiasis [21]. Additionally, the African Programme for Onchocerciasis Control (APOC) study was used to identify areas for community MDA with ivermectin and map loiasis [83].

**Development of bespoke integrated survey protocols.** While leveraging existing platforms was common, nesting or using existing single-disease protocols may not always be optimal for effective integration, especially when aiming for a balance between epidemiological rigour and field practicality. Some studies developed new integrated study protocols to address the complexities of combining multiple diseases [40,46,62,63]. These protocols are typically aimed at balancing epidemiological rigour with field practicality, and were developed with input from programme managers, key stakeholders, and disease-specific experts from the beginning to ensure that clear survey objectives, relevance to each disease, and alignment with international guidelines. The development of a new integrated protocol requires careful planning and should include input from specialists covering all targeted diseases to ensure that the survey addresses the specific needs of each condition while maintaining overall coherence and quality [93]. While developing new protocols is a more resource-intensive undertaking in terms of planning, expert input, and validation, it allows a more optimised and tailored solution.

## Enabling factors and opportunities for integration

**Overlapping disease distributions and implementation timelines.** Integration becomes more feasible and efficient when the targeted diseases share similar epidemiological characteristics. One of the primary drivers of integration is regional co-endemicity. Integrated M&E is logical and effective for NTDs with overlapping geographic distribution that have similar patterns of clustering by age and gender [82–85].

In addition, integration is facilitated by similarity in clinical manifestation and overlap in the target age group. For example, Sime *et al*. (2014) highlight that lymphoedema due to LF and podoconiosis has similar clinical manifestations and affects the same age group, making combined assessment sensible, especially since podoconiosis diagnosis requires exclusion of LF [91]. For diseases like scabies and impetigo, focusing on children can facilitate integration with surveys for STH, which are often conducted in schools. Integrating mapping of scabies with surveys for trachoma, maternal and child health services, vaccine coverage, or malaria may be practical, but challenges may arise when diseases require significantly different assessment procedures [82]. Similarities in the target age group allow for one technical team to efficiently collect data for multiple diseases by sampling the same individuals [85].

An important enabling factor for integration is the ability to use an existing program platform and infrastructure. Utilising the structure, resources, and logistics of an ongoing health program significantly reduces the need to build new infrastructure from scratch, saving resources and time [31,37,39,50,58,59,64,81,85,93,95]. Existing, funded NTD ventures, such as the standardised trachoma prevalence surveys [59] or Lymphatic Filariasis Transmission Assessment Surveys (TAS) [39,50], can serve as effective platforms for integrating the mapping of other NTDs like yaws, guinea worm, schistosomiasis, STH, and LF [82]. This "piggybacking" or "nesting" approach minimises overhead costs, reduces planning complexity, and optimises resource utilisation (e.g., shared transport, per diems, training), thereby enhancing practicality and cost-effectiveness.

Integration with other disease programs beyond NTDs, such as TB, malaria, HIV, and vaccination programs has been recommended, as has coordination with other platforms, such as cold chain and education [93,95]. Further suggestions include leveraging of existing population-based surveys, such as Demographic and Health Surveys (DHS), malaria and AIDS Indicator Surveys [89]. Cooley *et al*. (2021) embedded an integrated yaws and trachoma survey in a SMART nutritional survey [64]. Similarly, Grimes *et al*. (2017) conducted an integrated schistosomiasis and STH survey in schools supported by the World Food Program (WFP) [74]. Gunawarden *et al*. (2014) leveraged an existing structure of the Anti-Filariasis Campaign (AFC) for school access, community engagement (parent meetings), and blood sample collection for

LF in Sri Lanka. The partnership between the AFC and an academic team facilitated integration, as the university could provide laboratory facilities and trained staff for STH testing which the AFC lacked [50]. Demographic surveillance systems (DSS) were also used as a platform to concurrently collect data on schistosomiasis, STH, LF, and malaria in Kenya [37]; schistosomiasis and malaria in Angola [31]; and STH, schistosomiasis, and malaria in Côte d'Ivoire [58], effectively utilising an existing program infrastructure. The use of the DSS facilitated community mobilisation, identification of eligible participants, and increased survey efficiency [31].

**Programmatic considerations.** Broad programmatic goals and the stage of the disease control program are important to consider before deciding to integrate NTD surveys. Engelman (2016) emphasises that the specific NTD goals, i.e., control, elimination, or eradication, determine the survey design. Programs aiming for NTD elimination or eradication require more extensive efforts to reach all populations. In contrast, control-focused programs may initially prioritise areas that are more accessible when conducting a survey. The program stage (baseline mapping *versus* impact assessment) also affects feasibility. Therefore, it is essential to define the priorities and align programs that have similar priorities when considering integration [82].

## Costs and savings of integrated surveys

Integrated surveys are expected to have higher costs than single-disease surveys due to the need for additional personnel, among other resources. Knipes *et al.* (2017) estimated that the cost of a TAS-malaria evaluation was 15% higher than an LF-only evaluation, and a TAS-STH-malaria evaluation was 49% higher than an LF-only evaluation [65]. However, any additional costs less than 100% of the combined standalone studies imply a cost saving from integration.

Sime *et al.* (2014) reported a total saving of $1,132,473 by integrating national surveys for LF and podoconiosis. The total cost of the integrated study was $1,291,400, compared to an estimated cost of $1,212,209 for LF-only mapping and $1,211,664 for podoconiosis-only mapping [91]. A survey that integrated LF, trachoma, schistosomiasis, and STH using the Integrated Threshold Mapping (ITM) methodology showed a 31% overall cost saving in Mali and a 19% overall cost saving in Senegal compared to the standard, non-integrated WHO methodologies [79].

Overall, cost savings arise from integrated surveys due to the sharing of personnel and transportation costs, which typically represent the majority of survey budgets. Dorkenoo *et al.* (2012) reported that *per diem* pay accounted for 41% of the average survey cost, while transport costs constituted 35% [63]. Similarly, the cost of staff *per diems* was found to be 49% of the total cost for the schistosomiasis and STH survey in Namibia [62].

## Barriers and challenges to integration

The major challenge in integrating multiple NTD surveys is methodological incompatibilities. Differences in disease epidemiology, diagnostic requirements, and programmatic targets also pose significant challenges for integrating surveys.

**Sampling strategy and sample size.** Variations in target sample sizes, age groups, and sampling methods for different diseases pose significant barriers to integration [32,35,44,64,89,91–93,97]. Target sample sizes generally depend on disease prevalence, spatial distribution patterns, and thresholds for decision making relating to MDA [40]. Cooley *et al.* (2021) noted that sample sizes designed for trachoma surveillance may be inadequate for estimating the prevalence of other diseases, such as yaws [64]. Similarly, an integrated survey of LF and onchocerciasis done in Senegal using a convenience sampling method following the African Programme for Onchocerciasis Control (APOC) methodology did not achieve the sample size required for a conclusive LF TAS based on WHO criteria [32]. Target age groups vary depending on age-dependent exposure rates, the time taken to develop outcomes of interest, the population eligible for intervention, and whether the survey aims to detect baseline prevalence or assess programmatic impact [98]. For example, young children are at the highest risk of trachoma, so 1–9-year-olds are targeted for mapping surveys to estimate the prevalence of active trachoma. In contrast, adults aged 15 years and above are surveyed for trachomatous trichiasis, the late stage of the disease. Risk of schistosomiasis tends to peak at a slightly later age, so the target population is school-age children

who are also the main group targeted for MDA [99]. On the other hand, adults aged 20 years or older are surveyed in Onchocerciasis Elimination Mapping (OEM) epidemiological surveys due to their longer potential exposure to the parasite compared to children, which allows for greater sensitivity in detecting anti-*O. volvulus* antibody response. In contrast, Stop MDA surveys for onchocerciasis specifically target children 5–9 years of age, since antibody detection in this age group is indicative of active exposure to the parasite that occurred after the initiation of MDA [100] (Table 1).

Differences in sampling strategies are typically driven by disease epidemiology, as well as the survey goal. Wilson et al (2016) found that onchocerciasis survey methods—which typically employ convenience sampling in high-risk villages—may not be suitable for LF, which requires random sampling for TAS [32]. On the other hand, an integrated survey of LF and onchocerciasis in Sierra Leone noted that the use of LF TAS school-based cluster survey sampling strategy might have underestimated the true prevalence of onchocerciasis, where sampling from first-line villages is recommended for impact assessment [35].

While school-based surveys are more operationally convenient than community-based surveys, they may underestimate the actual community prevalence due to school attendance rates, sex bias, and failure to capture the target age group [42,57,65,78]. The target age groups for sampling differ among NTDs: school-aged children are targeted for schistosomiasis and STH surveys, children aged 6–7 years for LF, and adults for onchocerciasis [84].

**Diagnostic challenges.** Differences in diagnostic methodology were reported as another challenge that complicates integration. The use of different tests and diagnostic approaches that require different sample types, such as urine, stool, and blood, adds complexity and expenses to a survey [40,45,66,84,85]. The coordination of different sample types for the different diagnostic methods is a challenge for effective integration. Integrating NTDs that require laboratory sample collection and processing for diagnosis can consume more time than those that can be clinically diagnosed, impacting the feasibility and efficiency of integration [93].

**Logistical challenges.** The collection of different sample types that require different processing methods adds complexity and requires additional time and strategic coordination. Knipes (2017) reported limited access to cold chain for immunochromatographic card test cards and DBS storage to be challenges [65]. Combined mapping efforts can be more complex and require much more planning and coordination than single-disease surveys. In some cases, integration attempts might even be difficult and lead to reduced coverage for some diseases [85]. For instance, Ella et al

**Table 1. Target age groups required or recommended by WHO for Selected NTD surveys.**

| NTD | Survey Category | Primary Target Age Group |
|---|---|---|
| Lymphatic Filariasis | Transmission Assessment Surveys (TAS) [101] | Children aged 6–7 years |
| | Epidemiological Monitoring Survey (EMS)/ IDA impact survey [101] | Adults aged ≥ 20 years |
| | Integrated transmission assessment surveys (iTAS) [102] | Children aged 5–9 years |
| Onchocerciasis | onchocerciasis elimination mapping (OEM) [103] | Adults aged ≥ 20 years |
| | Stop-MDA Survey [103] | Children < 10 years |
| | Integrated transmission assessment surveys (iTAS) [102] | Children aged 5–9 years |
| Scabies | Prevalence Survey [104] | All age groups |
| Schistosomiasis | Baseline survey/ Impact assessment survey [105] | School-age children (5–14 years) |
| STH | Baseline survey/ Impact assessment survey [105] | School-age children (5–14 years) |
| Trachoma | Baseline survey/ Trachoma impact survey (TIS) [106] | Children 1–9 years & adults aged ≥ 15 years |
| | Trachomatous Trichiasis (TT) Survey [106] | adults aged ≥ 15 years |
| Yaws | Sero-survey [107] | Children aged 1–5 years |

(2022) noted that underestimating the number of devices needed per team and the human resources required led to a prolonged study time [20].

**Programmatic and organisational barriers.** Survey integration poses several programmatic and organisational challenges due to the complexities of aligning diverse program goals, differing methodological requirements, and organisational structures. Many NTD programs have disease-specific mapping guidelines. The absence of integrated mapping guidelines makes reaching consensus among different stakeholders on survey design and parameters challenging and time-consuming [91–93].

Working with different organisations further complicates planning and implementation [91]. Disease-specific elimination goals and funding programs are also barriers, as resources are often allocated for specific activities with differing reporting requirements, hindering compromise. Furthermore, there may be resistance to integration due to fear of loss of control, power or funding [93]. In addition, since integration is still a relatively new concept in many programs, hesitancy from field team members to work together can also be another challenge [63,79].

### Lessons learned and recommendations from an integrated survey experience

**Stakeholder collaboration.** Stakeholder collaboration at the national and district level, including government ministries, non-governmental organisations, and district health offices, was found to be essential for successful planning and implementation. Successful implementation requires high-level political and technical engagement, along with integrated planning. The formation of inter-programmatic and interdisciplinary work teams, guided by a clearly defined country coordinating team with diverse expertise, is crucial for designing and implementing an integrated mapping effort. This ensures that the diverse disease programs are aligned, protocols are harmonised, and data can be effectively shared and interpreted across diverse stakeholders, leading to a more cohesive national NTD strategy [79,91,92].

**Leveraging existing infrastructure and resources.** Leveraging existing health systems by maximising the use of pre-existing logistic frameworks, trained personnel, and community networks was demonstrated to be key to practical and effectively integrated surveys. Integration of NTDs into the national health information system for routine data collection and analysis is recommended. This integrated data platform may include joint administration of surveys for several NTDs and can strengthen data collection and reporting [95]. Utilising community health workers, who are already involved in different public health programs and have community acceptance, can be an effective way to conduct an integrated patient search in the community [43,78].

**Optimising methodology and data collection.** Refining diagnostic and data collection methods, utilising technology such as GIS and mobile technology, and adapting sampling methods are crucial for enhancing the accuracy, efficiency, and acceptability of integrated surveys. Multiplex antibody assays or portable diagnostic, self-contained platforms that are capable of simultaneously detecting antibodies to multiple pathogens are highly recommended for cost-effective surveillance and to reduce the number of surveys required [15,33,50,89].

Mobile technology facilitates data collection within large-scale integrated surveys by improving data quality through validation [92]. Geographical Information Systems (GIS) and remote-sensing techniques can support integrated M&E by enabling the identification of co-endemicity.

**Strategic planning and robust design.** Careful planning, informed by epidemiological and stakeholder input, is crucial for developing and implementing a scientifically sound and operationally feasible integrated survey. This includes identifying stakeholders, setting priorities, devising mitigation strategies, securing national government buy-in, defining clear objectives, developing local capacity, planning logistics, and obtaining necessary approvals [93].

Integration should be needs-based for all the programs involved. The survey design must be robust to address the objectives for each disease, so experts from all diseases should be included in the protocol design [92,93].

Survey strategies optimised for one disease may not be optimal for another, and there are limitations to integration [35,79]. It is essential to be flexible and to acknowledge and proactively address these potential limitations and challenges, such as inherent trade-offs in epidemiological rigour or logistical hurdles, to plan a robust and cost-effective integrated survey [44,50,54,56,57,62,65,79].

## Discussion

This scoping review systematically identified and synthesised existing evidence on methods and strategies for integrating population-based surveys for neglected tropical diseases (NTDs). We included 81 articles that conducted an integrated survey or discussed strategies and methods for integration. The NTDs most commonly targeted in integrated surveys were STH and schistosomiasis, aligning with qualitative findings that integration is facilitated by the overlap of sample types and target age groups. Other facilitators included multiplex diagnostics, the existence of established platforms for data collection, and programmatic willingness for integration. Integrated surveys were found to be significantly cost-effective, primarily by optimising shared resources, reducing redundant efforts, and minimising key cost drivers. The major expenditure for surveys was for field work, as staff *per diems* and transport costs accounted for a significant portion. By sharing the cost of field work, integrated surveys will only have an incremental cost due to additional training time and diagnostics.

The broad uptake of integrated surveys for NTDs aligns with WHO recommendations to shift from multiple disease-specific vertical programs to holistic, cross-cutting approaches. Integration is positioned by WHO as a critical mechanism to maximise the impact of and sustain national and global NTD control programs [108]. Integrated approaches facilitate the sharing of resources, personnel, and operational platforms, making integration a cost-efficient process that maximises the return on public health investment in low-resource countries. The countries with the highest numbers of published integrated surveys were Ethiopia, Côte d'Ivoire, Kenya, Nigeria, Sudan and Tanzania. As well as having high populations requiring interventions against NTDs, all of these countries were early adopters of integrated programmes, having been found to have integrated control for NTDs by 2018 [109].

The majority of integrated surveys we identified focused on PC NTDs, reflecting the need for population-based prevalence surveys to guide MDA decisions. In addition, the overlapping epidemiological distributions and the existing standardised survey methodologies for PC-NTDs make them well-suited for integrated surveys. Although integrated detection of skin NTDs was recommended by several of the review articles we identified, practical examples of integrated skin surveys were limited. Most skin NTDs are controlled primarily through individual-based management, except for LF—for which surveys use antigen prevalence—and scabies, for which control is not currently supported by large-scale donation programmes. Since MDA is recommended for scabies control by WHO [104], and many countries in Africa [110–112] have included targets for scabies control within their NTD Master Plans, the need for population-based prevalence surveys for scabies is likely to increase. In the absence of dedicated external funding for scabies control, programmes should seek opportunities to integrate scabies prevalence estimation within ongoing surveys for other outcomes.

Following the recognition of skin diseases as a global health priority by the World Health Assembly [113], scabies surveys could also offer a useful platform for integration of skin NTD diagnosis. However, programmes should consider the relative costs and benefits of integrating screening for other skin presenting NTDs within such surveys, based on co-endemicity, expected prevalence, and programmatic priorities to increase detection [83,114,115]. Skin NTDs controlled through individual based management usually occur at low prevalence, making prevalence estimation very costly, and their diagnosis is often complex. As such, skin examinations within scabies surveys may be viewed as an opportunity to integrate suspect case detection and referral of other skin NTDs, rather than prevalence estimation. This will rely on parallel strengthening of facility-based services for the diagnosis and management of skin diseases.

Utilising the existing community health workers (CHW) network was recommended as a strategy to integrate surveys, due to CHW community acceptance and existing involvement in public health programs. However, many studies show that CHWs already have a high workload due to multiple tasks [116]. Involving them in an integrated survey or patient identification may further increase this burden. To ensure effective integration, workloads should be clearly defined, manageable, and supported with adequate supervision and compensation.

Integrated NTD mapping was facilitated by the use of technologies such as mobile data collection platforms [92]. Emerging digital health technologies like mHealth (mobile health), eHealth, and telehealth have the potential to facilitate

remote data collection by allowing frontline health workers or community members to collect data directly in the community. This makes surveys more flexible by not requiring survey teams for data collection [117], making surveillance and mapping cost-effective.

Techniques to simultaneously detect multiple biomarkers, such as MBA, offer substantial efficiencies for integrated serosurveillance. However, MBA currently presents significant technical and logistical challenges, particularly in resource-limited settings. The development of these assays is technically demanding and requires extensive knowledge. Logistical and operational barriers, such as the need to maintain a strict cold chain for reagents and supplies is also a significant challenge [118].

Successful integration depends on multiple factors and is an iterative process that requires a holistic approach. Key recommendations include stakeholder and community engagement to build trust and collaboration, strategically identifying and collaborating with existing programs to ensure sustainability and efficiency, extensive planning informed by epidemiological data and stakeholders' input, and being adaptive and continuously optimising methodology. These support the planning and implementation of integrated surveys that are scientifically sound, practical and cost-effective. Community engagement and participation are fundamental for the success of an integrated survey. Early engagement with the community, before data collection starts, is essential for building trust, increasing participation, and preventing the spread of misinformation. Cha *et al.* (2017) reported parental resistance in some communities as a challenge, where parents were hesitant to allow their daughters to provide samples. This resistance was due to rumours that the nationwide survey aimed to harm community members or that the treatment would cause infertility in women [119]. Offering diagnosis and treatment to broader conditions and using less invasive diagnostic methods are likely to increase community participation [32,78]. Yotsu *et al.* (2018) found that integrating a survey with an MDA for STH and providing diagnosis and treatment for all skin conditions increased community acceptance [78].

Limited funding is a constant and growing concern for NTD control programs [120], which are mainly donor-funded. Many countries rely on external funders to sustain the progress in NTD reduction, making them vulnerable to shifts in donor priorities, jeopardising the sustainability and progress of the programs [121]. Recent global geopolitical shifts, causing reductions in foreign aid and the sudden termination of initiatives like USAID and programmes within the US Centres for Disease Control and Prevention, have created significant gaps in funding and coordination [122]. The identification of opportunities for integration should also be considered within broader health programmes seeking to optimise the use of scarce resources. However, the DHS and MIS programmes, which were recommended as a platform for integration, were terminated in 2025, and their future is uncertain [123]. These disruptions threaten to reverse decades of progress. This highlights the urgent need for an integrated framework to sustain the progress made so far and to achieve the goal of eliminating NTDs.

Target populations, survey methods and sample size are frequently mentioned barriers for integration. In this new landscape of scarcity, NTD programs, donors and WHO focal points should encourage optimising resources, which may necessitate compromise on certain disease-specific survey protocols. For example, if a trachoma impact survey is being conducted in an area historically endemic for LF, collecting data on LF will provide the program with valuable information, even if the target age groups aren't perfectly aligned. This is especially important when it comes to the context of post-validation surveillance-a time when funding for a vertical NTD program will be non-existent. Another challenge for integration is reporting. When integration leads to a slightly different age group or sampling method, NTD programs may find it challenging to report the results to WHO in the standard Epidemiological Data Reporting Form (EPIRF) template. Therefore, WHO should include more flexible reporting templates to encourage and enable integration.

This study is subject to some limitations. As a scoping review, it did not assess the quality of evidence or risk of bias in the included sources. In addition, only articles published in English were included due to time and resource constraints, missing the opportunity to include relevant evidence published in other languages. The review excluded sources for which

full-text articles could not be accessed, which may have omitted valuable insights available only in abstract form. While the inclusion of evidence from diverse global settings provided a broad perspective, the thematic analysis did not consider variations in geographical, healthcare system, or political contexts that may influence the feasibility and effectiveness of integration strategies.

## Conclusion

This scoping review found that strategies for integration revolve around leveraging existing infrastructure, identifying synergies between diseases based on shared characteristics, carefully harmonising methodologies and operational aspects, aligning with broader programmatic goals, and securing essential support. Integration should be based on strategic program alignment and co-endemicity of NTDs within countries for efficiency and sustainability. Successful integration is an iterative process that requires a holistic approach that combines sound epidemiological principles with practical field considerations, strong leadership, and community engagement.

## Supporting information

**S1 PRISMA Checklist. Scoping reviews (PRISMA-ScR) checklist.**
(PDF)

**S1 Appendix. Search strategy.**
(PDF)

**S2 Appendix. Data extraction form.**
(PDF)

**S3 Appendix. Extracted data.**
(XLSX)

## Acknowledgments

We thank Dr. Katherine Gass for providing useful feedback and suggestions on a draft of this manuscript.

## Author contributions

**Conceptualization:** Amanuel Belay, Hope Simpson.

**Data curation:** Amanuel Belay.

**Formal analysis:** Amanuel Belay.

**Funding acquisition:** Abebaw Fekadu, Gail Davey.

**Investigation:** Amanuel Belay.

**Methodology:** Amanuel Belay, Hope Simpson.

**Project administration:** Amanuel Belay.

**Supervision:** Agumasie Semahegn, Binyam Tesfaw Hailu, Abebaw Fekadu, Gail Davey, Hope Simpson.

**Validation:** Gail Davey, Rana Ahmed, Hope Simpson.

**Visualization:** Amanuel Belay.

**Writing – review & editing:** Amanuel Belay, Agumasie Semahegn, Binyam Tesfaw Hailu, Abebaw Fekadu, Gail Davey, Rana Ahmed, Hope Simpson.

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
