## [Decision Letter · Decision Letter 0]

5 Jan 2026

PNTD-D-25-02028

Integration of population-based surveys for Neglected Tropical Diseases: A Scoping Review

Dear Dr. Belay,

Thank you for submitting your manuscript to PLOS Neglected Tropical Diseases. After careful consideration, we feel that it has merit but does not fully meet PLOS Neglected Tropical Diseases's publication criteria as it currently stands. Therefore, we invite you to submit a revised version of the manuscript that addresses the points raised during the review process.

Please submit your revised manuscript within by Mar 06 2026 11:59PM. If you will need more time than this to complete your revisions, please reply to this message or contact the journal office at plosntds@plos.org. Please include the following items when submitting your revised manuscript:

We look forward to receiving your revised manuscript.

Kind regards,

Michael Marks

Academic Editor

Paul Brindley

Editor-in-Chief

Shaden Kamhawi

co-Editor-in-Chief

Paul Brindley

co-Editor-in-Chief

**Additional Editor Comments (if provided):**

**Journal Requirements:**

**Reviewers' Comments:**

Reviewer's Responses to Questions

**Key Review Criteria Required for Acceptance?**

**Methods**

-Are the objectives of the study clearly articulated with a clear testable hypothesis stated?

-Is the study design appropriate to address the stated objectives?

-Is the population clearly described and appropriate for the hypothesis being tested?

-Is the sample size sufficient to ensure adequate power to address the hypothesis being tested?

-Were correct statistical analysis used to support conclusions?

-Are there concerns about ethical or regulatory requirements being met?

Reviewer #1: Yes, yes, yes, yes, yes, no

Reviewer #2: The authors use standard methodology for their scoping review.

Reviewer #3: The inclusion criteria were not completely clear as laid out in the methods section. The authors state that they were targeting surveys with multiple NTDs. However, from a review of the reference section there were papers that were just one NTD along with malaria for example (reference 20). Further, how did they deal with multiple publications from the same set of surveys? For example, I am familiar with the schistosomiasis literature and was able to identify two studies that should have been considered in this scoping review. The first was a survey done in one zone of Amhara, Ethiopia, and the second was a set of surveys done in another nine zones of Amhara, Ethiopia, to complete the map. It seems like the second paper was dropped. It is not clear why.

King JD, et al. Intestinal parasite prevalence in an area of ethiopia after implementing the SAFE strategy, enhanced outreach services, and health extension program. PLoS Negl Trop Dis. 2013 Jun 6;7(6):e2223. doi: 10.1371/journal.pntd.0002223. PMID: 23755308; PMCID: PMC3675016.

Nute AW, et al. Prevalence of soil-transmitted helminths and Schistosoma mansoni among a population-based sample of school-age children in Amhara region, Ethiopia. Parasit Vectors. 2018 Jul 24;11(1):431.

**Results**

-Does the analysis presented match the analysis plan?

-Are the results clearly and completely presented?

-Are the figures (Tables, Images) of sufficient quality for clarity?

Reviewer #1: Yes, yes, yes

Reviewer #2: The review appears to be thorough.

Reviewer #3: The main addition that is needed for this manuscript is the table that describes the included studies and the relevant information that was extracted. This is commonly included in reviews such as this one. This should be a table of 81 rows and should report the information that was reviewed such as year of study, age range, type of assessments, NTDs, etc (i.e. the variables on the S3 data extraction form). This is listed as having been done in the PRISM document, but it is not present in this submission. The authors can include this as appendix or in the main text file, but it needs to be included so the reader knows which studies were detected. It is too difficult for the reader to parse through the results text and the reference section to piece together the nature of the studies included. Having this table available will be helpful to future researchers.

The authors rightly note that a key challenge to integrating surveys is that target age groups differ. It would be very helpful to the reader if the authors could include a table displaying the required or suggested age groups for the various surveys they reviewed. This would really help make the point about how survey design will be important to reach the required age groups.

**Conclusions**

-Are the conclusions supported by the data presented?

-Are the limitations of analysis clearly described?

-Do the authors discuss how these data can be helpful to advance our understanding of the topic under study?

-Is public health relevance addressed?

Reviewer #1: Yes, yes, yes, yes

Reviewer #2: See comments below.

Reviewer #3: It was often difficult to tell the difference between the results section and the discussion section. There were instances in the results where the authors moved away from presenting results and moved more towards commenting on the results. For example, see line 245-249 where the authors compare their approach with past approaches, something that would normally be found in the Discussion. Whereas in the Discussion lines 553-558, the authors introduce mHealth for the first time. This reads more like a finding or result of the scoping review and should be moved up into the results section.

The authors cover the MBA for serology in the results section. It would have been helpful to discuss the potential limitations of this approach in the discussion. Potentially in the paragraph around limited funding for NTD programs. These machines are expensive to keep supplied and to keep functioning. Second there are the decisions that go into which antigens to place on the beads, and those decisions are likely tied to where funding is coming from. Lastly, some of the antigens are still not completely agreed upon (see LF, Dengue). Some of these concerns should be raised so that there is a balanced assessment of the MDA approach.

The authors focus most of their manuscript on the PC-NTDs. This is likely justified given that most of the papers they found (per Fig 3) were among those diseases. However, it would be helpful to have their thoughts on how to integrate monitoring and evaluation or surveys for the other diseases. Is it that integration is not as important for those diseases, or is it that no one has tried? More discussion on this would be enlightening.

**Editorial and Data Presentation Modifications?**

Reviewer #1: Thank you for the opportunity to review this interesting paper. I have been trying to push for survey integration for many years. It is extremely difficult, as the authors note. I think this publication can help.

The authors should be congratulated for completing a big review and for the even analysis and nice presentation. My comments are minor.

39-40 : The Global Trachoma Mapping Project was a specific, time-limited project to map the baseline prevalence of signs of trachoma in populations in which this had not previously been done, funded by DFID and USAID. It finished in 2016. Suggest replace “and the Global Trachoma Mapping Project “ with “and standardized trachoma prevalence surveys”.

54: suggest change the word “regions” (which has political connotations in some countries), with “areas” or similar.

81-82: extremely pedantic point from me here. Suggest list the diseases either in alphabetical order or the order in which the donations were made (since this is a sentence about history) – if you choose the latter, oncho would come first, but I am not sure about the order of the others.

83-84: “Population-based prevalence surveys are … also used to monitor and evaluate the impact of control programs”. We use that language (about surveys evaluating impact) all the time, but in fact this is not really the role of repeated surveys, which is, in my view, always forward-facing – it’s to allow programmes to make decisions on what to do next, to stop or continue interventions. Evaluation of impact can only be done in a setting in which the simultaneous effect of other exposures can be controlled for.

87: This point will seem pedantic, but to me is extremely important. If we say, “Surveys drive up the cost of control and elimination programmes…” it suggests that surveys are somehow *outside the programme* and therefore optional extras or luxuries, perhaps done to satisfy the whims of academics or interested bystanders. In fact they are an essential part of the programme; MDA without surveys would be reckless, because the likely future benefit of the intervention would be unknown and the costs (money, time, adverse events, AMR) are non-zero. You have made the case for surveys in the previous para. Please don’t sweep that away here.

220: “two were a perspective article” – suggest edit to “two were perspective articles”

282: suggest change “collected” to “collecting”

289: I appreciate that this is a particular interest of the research group, but strictly speaking, podoconiosis is not an NTD in its own right, as least, not an NTD on the WHO list

295-298: “To support this, WHO has developed “Recognising neglected tropical diseases through changes on the skin’, a simple tool focusing on major skin changes such as ulcers, lumps, swollen limbs, and patches, and provides a diagnostic flowchart and algorithm for detecting and managing NTDs, can be used for mapping activities”. This sentence needs some editing to make complete sense (though the meaning is clear.) It might be easier to break it into shorter sentences.

298-299: “which was developed by WHO” is redundant here, the point having been noted in the previous sentence.

308-318: the Tropical Data collaboration, which took over trachoma prevalence survey support when the Global Trachoma Mapping Project finished and supports health ministries around the world to conduct several hundred district-level trachoma prevalence surveys each year, is potentially one of the most useful platforms for integration – because the trachoma survey process is rigorously quality controlled and quality assured, and now well-grooved. It would be worth mentioning here.

353: same comment as for 39-40 – the GTMP was funded, but is no longer “existing”. Please also see the comment above about Tropical Data, which may be worth referring to here instead of the GTMP

394: ITM stands for integrated threshold mapping

403: I think the major challenge to integration is lack of will to make it happen, but that’s a personal view that is probably not identifiable in the literature that you identified

418: please delete the word “infection” after “trachoma”

419: please change “measure” to “estimate” and delete “(follicular)”

420: please change “trachoma trichiasis” to “trachomatous trichiasis”

421: please change “schistosome infection” to “schistosomiasis”

423: please change “for” to “in”

437-438: also, kids going to school may be systematically different in terms of risk than non-school-going kids

547 and 549: CSW – should this be CHW?

568: * Recent global geopolitical shifts, such as reductions” – suggest change “such as” to “causing”

586-587: “generating some information is much better than generating no information”. Statements similar to this are often used to justify collecting bad data. I disagree with it *as a general statement*. When decisions for populations are made on the basis of data collected (and they are), we need to collectively make sure that the data address questions that are programmatically relevant, and are interpreted in the right way. (The example that you subsequently give in this para about LF seems sound, however.)

594: when abbreviated, WHO does not take the definite article. (The band does.) In other words, please delete the “the” in front of “WHO”.

Reviewer #2: I don't have have editorial corrections.

Reviewer #3: Abstract, line 39-the platform for trachoma is officially called “Tropical Data.” Suggest using this term throughout when referring to the ongoing trachoma surveys. GTMP was the original phase of baseline surveys, however, that project is now completed.

Introduction line 88- consider the use of the term “survey teams” instead of “health teams.”

Line 87, while surveys do certainly cost money, in the end they can be cost saving as they can inform a program to stop costly MDA and other interventions, right?

Methods lines 119-121 are confusing. The authors state that they present findings of the second research question only, however, they clearly cover the other 3 points throughout the manuscript, as they have figures showing which countries the work has been done in, and sections discussing challenges. Can the authors please clarify what is meant, or remove the sentence?

Lines 350-353 are very similar to lines 314-318 and make the same point with very similar wording. Please choose one location to make the point about the importance of having existing platforms.

Figure 2: the authors use the word publications in the title and the axes. Were the only papers identified those that were published? The search strategy included grey literature, no? If no grey literature articles were found, this should be reported.

**Summary and General Comments**

Reviewer #1: (No Response)

Reviewer #2: In this manuscript, Belay and colleagues present the results of a scoping review on integrated surveys for NTDs. Their review is thorough and the article is well written and includes a useful discussion of the barriers to integrating surveys. The authors also discuss the limitations of their review (e.g., the restriction to English-language articles).

My concerns about the paper are based on the dramatic shifts in the funding landscape that occurred in 2025. In the current context, it will be essential for the NTD community to radically change the way that M&E is done. In the face of funding shortfalls, disease-specific surveys will be very hard to justify – is there a donor that would pay $1M for an LF survey in one country at this point (e.g., line 390)? Integration of NTD surveys with malaria, VPD, and other surveys will be essential. These issues are noted in the text in a couple of places (e.g., a paragraph on pages 22-23), but these mentions feel like an afterthought. I think this paper would be much more useful to readers if integrated surveys – beyond just NTDs - received more attention as a necessity in the current context.

Reviewer #3: The manuscript describes a scoping review seeking to identify published manuscripts which describe integrated survey methodologies, including two or more NTDs. Integration will likely be a necessary step for governments and ministries of health going forward, and it helps to have a review of what has been done so far. The manuscript is generally well written, and the analysis is sound. There are a few issues that the authors should consider to improve the manuscript.

PLOS authors have the option to publish the peer review history of their article (what does this mean?). If published, this will include your full peer review and any attached files.

Reviewer #1: No

Reviewer #2: No

Reviewer #3: No

**Figure resubmission:**
While revising your submission, we strongly recommend that you use PLOS’s NAAS tool (https://ngplosjournals.pagemajik.ai/artanalysis) to test your figure files. NAAS can convert your figure files to the TIFF file type and meet basic requirements (such as print size, resolution), or provide you with a report on issues that do not meet our requirements and that NAAS cannot fix.
---

## [Editor Report · Decision Letter 1]

23 Mar 2026

Dear Mr Belay,

We are pleased to inform you that your manuscript 'Integration of population-based surveys for Neglected Tropical Diseases: A Scoping Review' has been provisionally accepted for publication in PLOS Neglected Tropical Diseases.

Best regards,

Paul J. Brindley, PhD

Editor-in-Chief

Paul Brindley

Editor-in-Chief

Shaden Kamhawi

co-Editor-in-Chief

Paul Brindley

co-Editor-in-Chief

---

## [Editor Report · Acceptance letter]

Dear Mr Belay,

We are delighted to inform you that your manuscript, "Integration of population-based surveys for Neglected Tropical Diseases: A Scoping Review," has been formally accepted for publication in PLOS Neglected Tropical Diseases.

Best regards,

Shaden Kamhawi

co-Editor-in-Chief

Paul Brindley

co-Editor-in-Chief
